# An Alternate Representation of the Vector of Apparent Power and Unbalanced Power in Three-Phase Electrical Systems

**Pedro A. Blasco**[ID]**, Rafael Montoya-Mira, José M. Diez ***[ID] **and Rafael Montoya**[ID]

Departamento de Ingeniería Eléctrica, Universitat Politècnica de València, Plaza Ferrandiz y Carbonell s/n, 03801 Alcoy, Spain; pedblaes@die.upv.es (P.A.B.); ramonmi@alumni.upv.es (R.M.-M.); rmontoya@die.upv.es (R.M.)

\* Correspondence: jmdiez@die.upv.es; Tel.: +34-9665-284-52

**Abstract:** Low-voltage distribution systems are typically unbalanced. These inefficiencies cause unbalanced powers that can significantly increase the apparent power of the system. Analysing and measuring these inefficient powers appropriately allows us to compensate for them and obtain a more efficient system. Correcting the imbalance at some nodes can worsen the rest of the system; therefore, it is essential that all nodes are analysed such that action can be taken when necessary. In most studies, the unbalanced power is measured from the modulus. Other more recent studies have proposed phasor expressions of unbalanced powers; however, in both cases, these are not enough to address the compensation of unbalanced powers in systems with unbalanced voltages. In this work, a different representation of the vector expressions for analysis of the unbalanced powers and the apparent powers of the three-phase linear systems is proposed. Additionally, these vector expressions are extended to nonlinear systems to quantify the harmonic apparent powers. These expressions have been formulated from the power of Buchholz and are valid for systems with unbalanced voltages and currents. To help understand the use of the proposed formulation, a practical case of a three-phase four-wire system with unbalanced loads and voltages is demonstrated.

**Keywords:** unbalanced power; power theory; apparent power; power system; power quality

## 1. Introduction

The majority of low-voltage distribution systems are unbalanced, mainly owing to the use of single-phase loads. In transmission systems or high-voltage systems, these effects are also present but they are very small. The phenomenon of imbalance causes unbalanced powers. The unbalanced powers constitute inefficiencies in the system [1–4] that cause increased losses in the lines and the malfunction of motors, generators, transformers, and protective equipment. In addition, the apparent power of the system is significantly increased [5–8]. Therefore, the powers due to imbalances must be calculated properly to facilitate the reliable design of active, passive, or hybrid devices, which can be used to compensate for them.

At present, there is no theory agreed on by all that allows an adequate evaluation of this type of phenomenon [9]. Most of the studies that exist focus on using the expressions proposed in the standard IEEE Std. 1459-2010 [10] and Buchholz [11]. These expressions use the Root Mean Square (RMS) values of the voltage and current that are expressed in symmetric or phase components. As a result, the moduli or RMS values of the apparent power and unbalanced power are obtained at a bus in the system [12]. These values are perfectly valid, but they are insufficient for accurate evaluation and analysis of the causes of the imbalance, as well as the place where they occur. In certain cases, the imbalances at a node

of the system can help to compensate for some of the imbalances at another node [13,14]. This type of case cannot be analysed correctly if we only know the moduli of the unbalanced powers, for example, when one knows the individual unbalance powers of two loads connected in parallel and one wants to calculate the unbalanced power that results from the coupling since the resulting unbalanced apparent power is not the arithmetic sum of both.

There are other studies that have proposed phasor or vector expressions to evaluate these inefficiencies. In [15], vector apparent power is defined as the vector sum of total active, total reactive, and total distortion powers, which are the sum of respective values for each phase. It treats each phase as an independent single-phase system like arithmetic apparent power. In this study, the unbalance power is not proposed by means of a vector expression. In [16], the unbalanced power vectors due to voltage and current are defined. To consider the unbalanced voltages, they use the unbalance factors $\delta_- = V_-/V_+$ and $\delta_0 = V_0/V_+$. In [17], the authors propose a new tool to phasorically represent the powers in the Clifford geometric space that is only applied to single-phase systems. In [18], the phasor of the total unbalance power is defined. It presents the same problem as [16] since they use the unbalance factors $\delta_-$ and $\delta_0$ to consider the unbalance power caused by the unbalanced voltages. Other authors extend their studies to the analysis of non-linear three-phase systems [19,20].

The use of these expressions significantly improves the capacity in the analysis of unbalanced phenomena and adapts correctly to systems fed with balanced voltages. Its application in systems with unbalanced voltages is based on including scalar expressions, such as factors $\delta_-$ and $\delta_0$. These scalar values diminish the capacity of analysis of vector expressions, making necessary their determination at any node of the system as we cannot use the vector properties that relate one node to another. Therefore, the unbalanced powers caused by the asymmetry of the voltages are not adequately considered. For this reason, when the voltages are unbalanced, it is also not possible to determine whether the compensation of the unbalanced powers at one node of the system improves or worsens the rest of the system.

The main objective of this paper is to propose the vector expressions necessary to quantify the unbalanced powers and, as a consequence, the total apparent powers of an unbalanced three-phase linear system. These expressions are valid for perfectly analysing systems with balanced and unbalanced voltages y, and in both cases, it allows us to develop passive compensators for unbalance powers due to negative and zero-sequence currents [21,22]. The modulus of the total apparent power obtained at any node in the system coincides with the apparent power of Buchholz. This value determines the maximum transferable power for each voltage waveform [23], including all efficient and non-efficient powers [24]. The components of these vectors are defined from terms of active and reactive power per phase; in this way, the same component of the vectors between two system buses or between loads connected in parallel can be arithmetically added to each other. From these expressions, it is possible to mainly analyze the unbalanced power flows in any electrical system. This allows, facing a change or variation in the system, analyzing the positive or negative contribution that this variation represents on the unbalance power and, as a consequence, its repercussion on the total apparent power on any bus of the system. Accordingly, in Section 2, based on the Buchholz power, the apparent power and unbalanced power of any system are analysed and studied for both balanced and unbalanced voltages. In Section 3, the new vector expressions of the apparent power are formulated. In Section 4, from the expressions deduced in the previous section, vector expressions of unbalanced power are formulated. In Section 5, the two typical cases to be considered in the use of the vectors of the apparent power and the vectors of the unbalanced power are studied: application to a bus with loads connected in parallel and application between two buses of the system united by a line. In Section 6, the formulation of the apparent power vector is extended to non-linear systems to determine the harmonic apparent power of the system. To help understand the use of these vector expressions, a practical case of a three-phase four-wire system with unbalanced loads and voltages is discussed in Section 7.

## 2. Analysis of the Apparent Power of Buchholz

Figure 1 shows a three-phase system with an unbalanced linear load powered by any sinusoidal voltage source. Considering the values of the voltages expressed in symmetrical components, this system can be divided into a total of three systems: the first caused by the positive-sequence voltage (see Figure 2), the second caused by the negative-sequence voltage (see Figure 3), and the third caused by the zero-sequence voltage (see Figure 4). The relationship between the impedances of the system and the three equivalent systems is given by (1).

$$
\begin{aligned}
\overrightarrow{Z_a} &= \overrightarrow{Z_{a+}} + \overrightarrow{Z_{a-}} + \overrightarrow{Z_{a0}} \\
\overrightarrow{Z_b} &= \overrightarrow{Z_{b+}} + \overrightarrow{Z_{b-}} + \overrightarrow{Z_{b0}} \\
\overrightarrow{Z_c} &= \overrightarrow{Z_{c+}} + \overrightarrow{Z_{c-}} + \overrightarrow{Z_{c0}}
\end{aligned}
\tag{1}
$$

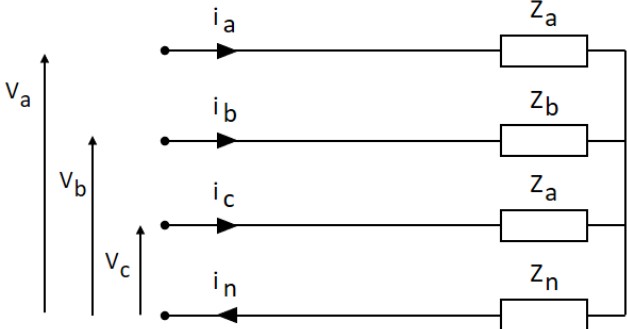

**Figure 1.** Electric system with unbalanced three-phase load and sinusoidal voltage source.

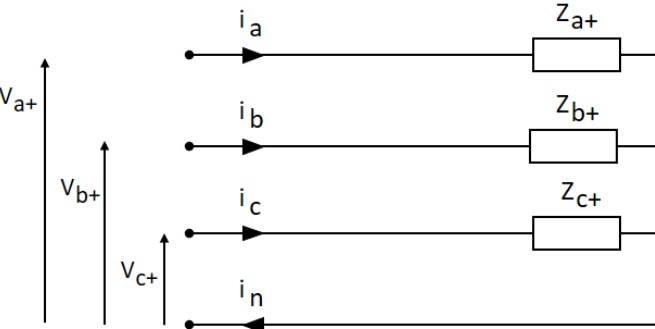

**Figure 2.** Equivalent system with positive-sequence voltage.

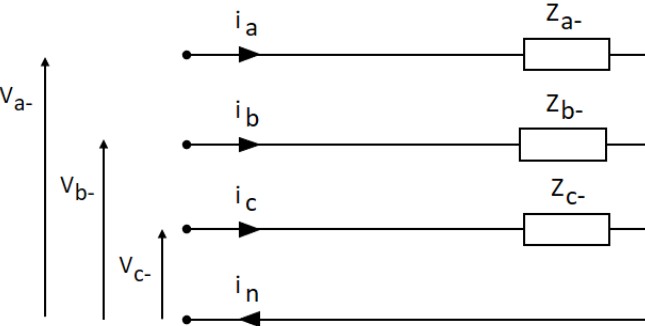

**Figure 3.** Equivalent system with negative-sequence voltage.

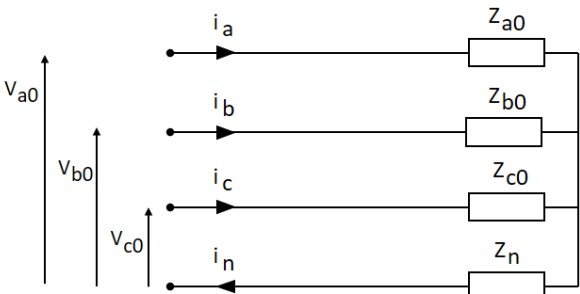

**Figure 4.** Equivalent system with zero-sequence voltage.

If we express the values of the voltages and currents in symmetrical components, the apparent power of Buchholz [11] of the system shown in Figure 1 is given by:

$$S_1 = 3 \sqrt{\left(V_+^2 + V_-^2 + V_0^2\right)\left(I_+^2 + I_-^2 + I_0^2\right)}$$
(2)

where

- $S_1$ is the total apparent power of the system.
- $V_+$, $V_-$, and $V_0$ are the *RMS* values of positive sequence, negative sequence, and zero-sequence voltages, respectively.
- $I_+$, $I_-$, and $I_0$ are the *RMS* values of positive sequence, negative sequence, and zero sequence currents, respectively.

By expanding (2), the following equation is obtained [10]

$$S_1{}^2 = S_{v+}^2 + S_{v-}^2 + S_{v0}^2$$
(3)

$$S_{v+} = \sqrt{9\, V_+^2\left(I_+^2 + I_-^2 + I_0^2\right)}$$
(4)

$$S_{v-} = \sqrt{9\, V_-^2\left(I_+^2 + I_-^2 + I_0^2\right)}$$
(5)

$$S_{v0} = \sqrt{9\, V_0^2\left(I_+^2 + I_-^2 + I_0^2\right)}$$
(6)

where

- In the first addend, $S_{v+}$ is the apparent power of the system caused by the positive-sequence voltage. Its modulus is given by (4), and its equivalent circuit is shown in Figure 2.
- In the second addend, $S_{v-}$ is the apparent power of the system caused by the negative-sequence voltage. Its modulus is given by (5), and its equivalent circuit is shown in Figure 3.
- In the third addend, $S_{v0}$ is the apparent power of the system caused by the zero-sequence voltage. Its modulus is given by (6), and its equivalent circuit is shown in Figure 4.

*2.1. Three-Phase Sinusoidal System with Unbalanced Loads and Balanced Voltages*

Under these conditions, the negative-sequence and zero-sequence voltages are zero. Therefore, (5) and (6) have null values, and the total apparent power of the system is given by (7).

$$S_1{}^2 = S_{v+}^2 = 9\, V_+^2\left(I_+^2 + I_-^2 + I_0^2\right)$$
(7)

Using the Fortescue transformation matrix, the line currents moduli in terms of symmetric components are expressed by (8)–(10).

$$
\begin{aligned}
|I_+|^2 &= \tfrac{1}{9}[I_a \cos\beta_a + I_b \cos(\beta_b + 120) + I_c \cos(\beta_c + 240)]^2 \\
&+ \tfrac{1}{9}[I_a \sin\beta_a + I_b \sin(\beta_b + 120) + I_c \sin(\beta_c + 240)]^2
\end{aligned}
\tag{8}
$$

$$
\begin{aligned}
|I_-|^2 &= \tfrac{1}{9}[I_a \cos\beta_a + I_b \cos(\beta_b + 240) + I_c \cos(\beta_c + 120)]^2 \\
&+ \tfrac{1}{9}[I_a \sin\beta_a + I_b \sin(\beta_b + 240) + I_c \sin(\beta_c + 120)]^2
\end{aligned}
\tag{9}
$$

$$
|I_0|^2 = \frac{1}{9}[I_a \cos\beta_a + I_b \cos\beta_b + I_c \cos\beta_c]^2 + \frac{1}{9}[I_a \sin\beta_a + I_b \sin\beta_b + I_c \sin\beta_c]^2
\tag{10}
$$

Substituting the sequence values from (8), (9), and (10) into (7), the following expression is obtained

$$
S_{v+}^2 = 3\, V_+^2\left(I_a^2 + I_b^2 + I_c^2\right)
\tag{11}
$$

Considering (11) and taking into account $V_+ = V_{a+} = V_{b+} = V_{c+}$, we get

$$
S_{v+}^2 = 3\left(V_{a+}^2 I_a^2 + V_{b+}^2 I_b^2 + V_{c+}^2 I_c^2\right)
\tag{12}
$$

The three addends inside the parenthesis of (12) represent the apparent powers in each phase of the system. Their values can be determined from the active and reactive powers, considering the following expressions [10]:

$$
\left(S_{a+}^a\right)^2 = \left(P_{a+}^a\right)^2 + \left(Q_{a+}^a\right)^2
\tag{13}
$$

$$
\left(S_{b+}^b\right)^2 = \left(P_{b+}^b\right)^2 + \left(Q_{b+}^b\right)^2
\tag{14}
$$

$$
\left(S_{c+}^c\right)^2 = \left(P_{c+}^c\right)^2 + \left(Q_{c+}^c\right)^2
\tag{15}
$$

where

- $P_{a+}^a$, $P_{b+}^b$, and $P_{c+}^c$ are the active powers due to the positive-sequence voltage in each phase; see Figure 2. Their values according to Steinmetz are calculated from (16).
- $Q_{a+}^a$, $Q_{b+}^b$, and $Q_{c+}^c$ are the reactive powers due to the positive-sequence voltage in each phase; see Figure 2. Their values according to Steinmetz are calculated from (17).

$$
P_{z+}^z = V_{z+} I_z \cos(\alpha_{z+} - \beta_z) \quad z = \{a, b, c\}
\tag{16}
$$

$$
Q_{z+}^z = V_{z+} I_z \sin(\alpha_{z+} - \beta_z) \quad z = \{a, b, c\}
\tag{17}
$$

Here, $\alpha_{z+}$ is the angle of positive-sequence voltages in each phase and $\beta_z$ is the angle of line currents in each phase.

Substituting (13) and (14) and (15) into (12), we get

$$
S_{v+}^2 = 3\left[\sum\left(P_{z+}^z\right)^2 + \sum\left(Q_{z+}^z\right)^2\right] \quad z = \{a, b, c\}
\tag{18}
$$

Therefore, when a system is powered by a balanced voltage source, the apparent power modulus $S_1$ is equal to $S_{v+}$ and is given by (19).

$$
S_{v+} = \sqrt{3}\sqrt{\sum\left(P_{z+}^z\right)^2 + \sum\left(Q_{z+}^z\right)^2} \quad z = \{a, b, c\}
\tag{19}
$$

*2.2. Three-Phase Sinusoidal System with Unbalanced Loads and Unbalanced Voltages*

When the voltages are unbalanced, in a three-phase, four-wire system, as shown in Figure 1, the values of the negative-sequence and zero-sequence voltages are not zero; therefore, the apparent

total power S will be determined by the three terms $S_{v+}$, $S_{v-}$, and $S_{v0}$. Their values are calculated from (3).

If for $S_{v-}$ and $S_{v0}$ we follow an approach analogous to the one made in the previous section for $S_{v+}$, the following expressions can be obtained:

$$S_{v-}^2 = 3\left[\sum (P_{z-}^z)^2 + \sum (Q_{z-}^z)^2\right] \quad z = \{a, b, c\} \tag{20}$$

$$S_{v0}^2 = 3\left[\sum \left(P_{z0}^z\right)^2 + \sum \left(Q_{z0}^z\right)^2\right] \quad z = \{a, b, c\} \tag{21}$$

where its moduli are given by (22) and (23).

$$S_{v-} = \sqrt{3}\sqrt{\sum (P_{z-}^z)^2 + \sum (Q_{z-}^z)^2} \quad z = \{a, b, c\} \tag{22}$$

$$S_{v0} = \sqrt{3}\sqrt{\sum \left(P_{z0}^z\right)^2 + \sum \left(Q_{z0}^z\right)^2} \quad z = \{a, b, c\} \tag{23}$$

where

- $P_{a-}^a$, $P_{b-}^b$, and $P_{c-}^c$ are the active powers due to the negative-sequence voltage in each phase; see Figure 3. Their values are calculated from (24).
- $Q_{a-}^a$, $Q_{b-}^b$, and $Q_{c-}^c$ are the reactive powers due to the negative-sequence voltage in each phase; see Figure 3. Their values are calculated from (25).
- $P_{a0}^a$, $P_{b0}^b$, and $P_{c0}^c$ are the active powers due to the zero-sequence voltage in each phase; see Figure 4. Their values are calculated from (26).
- $Q_{a0}^a$, $Q_{b0}^b$, and $Q_{c0}^c$ are the reactive powers due to the zero-sequence voltage in each phase; see Figure 4. Their values are calculated from (27).

$$P_{z-}^z = V_{z-}I_z \cos(\propto_{z-} -\beta_z) \quad z = \{a, b, c\} \tag{24}$$

$$Q_{z-}^z = V_{z-}I_z \sin(\propto_{z-} -\beta_z) \quad z = \{a, b, c\} \tag{25}$$

$$P_{z0}^z = V_{z0}I_z \cos(\propto_{z0} -\beta_z) \quad z = \{a, b, c\} \tag{26}$$

$$Q_{z0}^z = V_{z0}I_z \sin(\propto_{z0} -\beta_z) \quad z = \{a, b, c\} \tag{27}$$

where $\propto_{z-}$ is the angle of negative-sequence voltages in each phase and $\propto_{z0}$ is the angle of zero-sequence voltages in each phase.

## 3. Apparent Power Vectors Proposed

As described in the previous sections, from the symmetric components of voltages, any unbalanced linear three-phase system is divided into three linear three-phase systems. $S_{v+}$, $S_{v-}$, and $S_{v0}$ are the apparent powers in each of these three systems, and their RMS values are calculated from (19), (22), and (23). Each of these expressions can be expressed in vector form three apparent power vectors: positive-sequence voltage apparent power vector $\overrightarrow{S_{v+}}$, negative-sequence voltage apparent power vector $\overrightarrow{S_{v-}}$, and zero-sequence voltage apparent power vector $\overrightarrow{S_{v0}}$. Each of these vectors has six components that are orthogonal to each other and are determined from the following expressions:

$$\overrightarrow{S_{v+}} = \sqrt{3}\left(P_{a+}^a\overrightarrow{u_{pa+}} + P_{b+}^b\overrightarrow{u_{pb+}} + P_{c+}^c\overrightarrow{u_{pc+}} + Q_{a+}^a\overrightarrow{u_{qa+}} + Q_{b+}^b\overrightarrow{u_{qb+}} + Q_{c+}^c\overrightarrow{u_{qc+}}\right) \tag{28}$$

$$\overrightarrow{S_{v-}} = \sqrt{3}\left(P_{a-}^a\overrightarrow{u_{pa-}} + P_{b-}^b\overrightarrow{u_{pb-}} + P_{c-}^c\overrightarrow{u_{pc-}} + Q_{a-}^a\overrightarrow{u_{qa-}} + Q_{b-}^b\overrightarrow{u_{qb-}} + Q_{c-}^c\overrightarrow{u_{qc-}}\right) \tag{29}$$

$$\overrightarrow{S_{v0}} = \sqrt{3}\left(P_{a0}^a\overrightarrow{u_{pa0}} + P_{b0}^b\overrightarrow{u_{pb0}} + P_{c0}^c\overrightarrow{u_{pc0}} + Q_{a0}^a\overrightarrow{u_{qa0}} + Q_{b0}^b\overrightarrow{u_{qb0}} + Q_{c0}^c\overrightarrow{u_{qc0}}\right) \tag{30}$$

where

- $\overrightarrow{u_{pz+}}$ and $\overrightarrow{u_{qz+}}$ are the unit vectors that are associated with the six components of the apparent power vector due to the positive-sequence voltage. These unit vectors are perpendicular to each other.
- $\overrightarrow{u_{pz-}}$ and $\overrightarrow{u_{qz-}}$ are the unit vectors that are associated with the six components of the apparent power vector due to the negative-sequence voltage. These unit vectors are perpendicular to each other.
- $\overrightarrow{u_{pz0}}$ and $\overrightarrow{u_{qz0}}$ are the unit vectors that are associated with the six components of the apparent power vector due to the zero-sequence voltage. These unit vectors are perpendicular to each other.

The total apparent power according to (3) is calculated from the vector moduli $\overrightarrow{S_{v+}}$, $\overrightarrow{S_{v-}}$, and $\overrightarrow{S_{v0}}$.

Moreover, active power $P$ and reactive power $Q$, used in the classical theories, are determined by the arithmetic sum of all the components of their nature, their expressions being the following:

$$P = \sum P_{z+}^z + \sum P_{z-}^z + \sum P_{z0}^z \quad z = \{a, b, c\} \tag{31}$$

$$Q = \sum Q_{z+}^z + \sum Q_{z-}^z + \sum Q_{z0}^z \quad z = \{a, b, c\} \tag{32}$$

## 4. Unbalanced Power Vectors Proposed

In a linear three-phase system balanced in voltages and currents, the negative sequence and zero sequence currents and voltages are null. Under these conditions, the total apparent power of the system is determined exclusively from the positive-sequence currents and voltages. This apparent power is called the apparent positive-sequence power $S_+$. Positive-sequence active power $P_+$ and positive-sequence reactive power $Q_+$ are calculated from (16) and (17).

If we consider these powers in each phase, the following conditions are satisfied:

$$P_{a+}^{a+} = P_{b+}^{b+} = P_{c+}^{c+} = \frac{P_+}{3} \tag{33}$$

$$Q_{a+}^{a+} = Q_{b+}^{b+} = Q_{c+}^{c+} = \frac{Q_+}{3} \tag{34}$$

In an unbalanced linear three-phase system [10], the only apparent power that is not due to the imbalance of the voltages and currents is the positive sequence apparent power $S_+$; the rest of the apparent powers are caused by imbalances, representing the unbalanced apparent power. As in the previous section, unbalanced power is defined from three vectors: vector of unbalanced apparent power of positive-sequence voltage $\overrightarrow{S_{uv+}}$, vector of unbalanced apparent power of negative-sequence voltage $\overrightarrow{S_{uv-}}$, and vector of unbalanced apparent power of zero-sequence voltage $\overrightarrow{S_{uv0}}$. The vectors $\overrightarrow{S_{uv-}}$ and $\overrightarrow{S_{uv0}}$, according to (29) and (30), represent only the unbalanced powers; therefore, $\overrightarrow{S_{uv-}} = \overrightarrow{S_{v-}}$ and $\overrightarrow{S_{uv0}} = \overrightarrow{S_{v0}}$. The vector $\overrightarrow{S_{v+}}$ includes $\overrightarrow{S_+}$, which is not an unbalanced power; therefore, to obtain $\overrightarrow{S_{uv+}}$, the corresponding values of $P_+$ and $Q_+$, which according to (33) and (34) are equal to one-third of its value, must be subtracted in each of its components, as in (35), where $z = \{a, b, c\}$.

$$\overrightarrow{S_{uv+}} = \sqrt{3}\left[\sum\left(P_{z+}^z - \frac{P_+}{3}\right)\overrightarrow{u_{pz+}} + \sum\left(Q_{z+}^z - \frac{Q_+}{3}\right)\overrightarrow{u_{qz+}}\right] \tag{35}$$

The modulus of unbalanced apparent power is given by

$$S_{uv+} = \sqrt{3}\sqrt{\sum\left(P_{z+}^z - \frac{P_+}{3}\right)^2 + \sum\left(Q_{z+}^z - \frac{Q_+}{3}\right)^2} \tag{36}$$

The total unbalanced apparent power $S_u$ according to (37) is calculated from the vector moduli $\overrightarrow{S_{uv+}}$, $\overrightarrow{S_{uv-}}$, and $\overrightarrow{S_{uv0}}$.

$$S_u = \sqrt{S_{uv+}{}^2 + S_{uv-}{}^2 + S_{uv0}{}^2} \tag{37}$$

Therefore, the apparent total power is given by (38).

$$S_1 = \sqrt{S_+{}^2 + S_{uv+}{}^2 + S_{uv-}{}^2 + S_{uv0}{}^2} \tag{38}$$

Additionally, if we express the line currents in symmetric components, the total apparent power is determined by (39).

$$S_1 = \sqrt{S_+{}^2 + S_+^{-2} + S_+^{0\,2} + S_-{}^2 + S_-^{+2} + S_-^{0\,2} + S_0{}^2 + S_0^{+2} + S_0^{-2}} \tag{39}$$

Following the same procedure as in previous sections, the apparent powers of any of the nine addends of the square root can be expressed by means of its vector expression according to (40) and its moduli from (41). Here $w = \{+, -, 0\}$ indicates the symmetric component of the voltage and $x = \{+, -, 0\}$ indicates the symmetric component of the current. When the symmetrical components of voltage and current are the same in (39), we have indicated it only with the subscript, therefore in (40) and (41); $S_+ = S_+^+$, $S_- = S_-^-$ and $S_0 = S_0^0$. On the other hand, $\overrightarrow{u_{pzw}^{zx}}$ and $\overrightarrow{u_{qzw}^{zx}}$ are the unit vectors associated with each active and reactive component, respectively.

$$\overrightarrow{S_w^x} = \sqrt{3}\left[\sum P_{zw}^{zx}\,\overrightarrow{u_{pzw}^{zx}} + \sum Q_{zk}^{zx}\,\overrightarrow{u_{qzw}^{zx}}\right] \tag{40}$$

$$S_w^x = \sqrt{3}\ \sqrt{\sum (P_{zw}^{zx})^2 + \sum (Q_{zw}^{zx})^2} \tag{41}$$

With these vectors, it is easy to analyse the electrical systems knowing the individual vectors in each of the loads. In the practical case of Section 7, the simplicity in the use of the proposed vectors is observed.

Furthermore, the decomposition of the apparent powers in different sequences helped the authors to obtain a passive compensator for the negative sequence current [21], treating the reactive power $Q_{z+}^{z-}$ in a similar way to the compensation of the positive-sequence reactive power. It was also applied to the development of a passive compensator for the zero-sequence current consumed by the load [22]. These compensators respond correctly for both balanced and unbalanced voltages.

## 5. Application of the Apparent Power Vectors and Unbalanced Power Vectors in a Three-Phase System

### 5.1. Loads Connected in Parallel on a System Bus

Figure 5 shows two linear three-phase loads in a star configuration in which they are connected to bus "*i*" of an electric power system. Under these conditions, the apparent power vectors on the bus are determined by (42)–(44). Each of the components of the resulting vectors on bus "*i*" is calculated from the homologous components of the vectors in each of the loads by means of their arithmetic sum.

$$\overrightarrow{S_{v+(bus\ i)}} = \overrightarrow{S_{v+(load\ 1)}} + \overrightarrow{S_{v+(load\ 2)}} \tag{42}$$

$$\overrightarrow{S_{v-(bus\ i)}} = \overrightarrow{S_{v-(load\ 1)}} + \overrightarrow{S_{v-(load\ 2)}} \tag{43}$$

$$\overrightarrow{S_{v0(bus\ i)}} = \overrightarrow{S_{v0(load\ 1)}} + \overrightarrow{S_{v0(load\ 2)}} \tag{44}$$

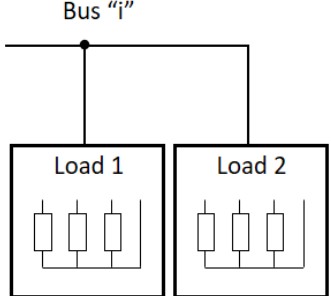

**Figure 5.** Two parallel linear loads connected to a system bus.

For unbalanced power vectors,

$$\overrightarrow{S_{uv+(bus\ i)}} = \overrightarrow{S_{uv+(load\ 1)}} + \overrightarrow{S_{uv+(load\ 2)}} \tag{45}$$

$$\overrightarrow{S_{uv-(bus\ i)}} = \overrightarrow{S_{v-(bus\ i)}} \tag{46}$$

$$\overrightarrow{S_{uv0(bus\ i)}} = \overrightarrow{S_{v0(bus\ i)}} \tag{47}$$

where for $z = \{a, b, c\}$ and $w = \{+, -, 0\}$,

$$P^z_{zw(bus\ i)} = P^z_{zw(load\ 1)} + P^z_{zw(load\ 2)} \tag{48}$$

$$Q^z_{zw(bus\ i)} = Q^z_{zw(load\ 1)} + Q^z_{zw(load\ 2)} \tag{49}$$

and also

$$P_{+(bus\ i)} = P_{+(load\ 1)} + P_{+(load\ 2)} \tag{50}$$

$$Q_{+(bus\ i)} = Q_{+(load\ 1)} + Q_{+(load\ 2)} \tag{51}$$

### 5.2. Two Buses of a System Linked by a Power Line

Figure 6 shows two buses in a system ($i$ and $j$) linked by a power line with impedance $Z_{ij}$. Under these conditions, the apparent power vectors in the line are determined by subtracting their corresponding homologous vectors in each of the buses according to (52)–(54).

$$\overrightarrow{S_{v+(line)}} = \overrightarrow{S_{v+(bus\ i)}} - \overrightarrow{S_{v+(bus\ j)}} \tag{52}$$

$$\overrightarrow{S_{v-(line)}} = \overrightarrow{S_{v-(bus\ i)}} - \overrightarrow{S_{v-(bus\ j)}} \tag{53}$$

$$\overrightarrow{S_{v0(line)}} = \overrightarrow{S_{v0(bus\ i)}} - \overrightarrow{S_{v0(bus\ j)}} \tag{54}$$

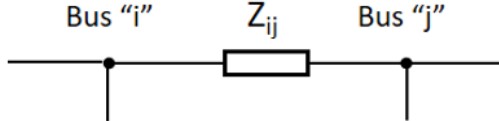

**Figure 6.** Two system buses linked by a power line

For unbalanced power vectors,

$$\overrightarrow{S_{uv+(line)}} = \overrightarrow{S_{uv+(bus\ i)}} - \overrightarrow{S_{uv+(bus\ j)}} \tag{55}$$

$$\overrightarrow{S_{uv-(line)}} = \overrightarrow{S_{uv-(bus\ i)}} - \overrightarrow{S_{uv-(bus\ j)}} \tag{56}$$

$$\overrightarrow{S_{v0(line)}} = \overrightarrow{S_{v0(bus\ i)}} - \overrightarrow{S_{v0(bus\ j)}} \tag{57}$$

where, for $z = \{a, b, c\}$ and $w = \{+, -, 0\}$,

$$P^z_{zw(line)} = P^z_{zw(bus\ i)} - P^z_{zw(bus\ j)} \tag{58}$$

$$Q^z_{zw(line)} = Q^z_{zw(bus\ i)} - Q^z_{zw(bus\ j)} \tag{59}$$

and also

$$P_{+(line)} = P_{+(bus\ i)} - P_{+(bus\ j)} \tag{60}$$

$$Q_{+(line)} = Q_{+(bus\ i)} + Q_{+(bus\ j)} \tag{61}$$

The quantities $P^z_{zw(line)}$, $Q^z_{zw(line)}$, $P_{+(line)}$, and $Q_{+(line)}$ can also be calculated from the impedances of the power line that exists between the two buses. Decomposing $Z_{ij}$ from (1) according to the positive-, negative-, and zero-sequence voltages: $Z_{z+}$, $Z_{z-}$ and $Z_{z0}$, obtaining the following expressions:

$$P^z_{zw(line)} = R_{zw}\ I^2_z \quad z = \{a, b, c\} \quad w = \{+, -, 0\} \tag{62}$$

$$Q^z_{zw(line)} = X_{zw}\ I^2_z \quad z = \{a, b, c\} \quad w = \{+, -, 0\} \tag{63}$$

$$P_{+(line)} = \sum R_{z+}\ I^2_{z+} \quad z = \{a, b, c\} \tag{64}$$

$$Q_{+(line)} = \sum X_{z+}\ I^2_{z+} \quad z = \{a, b, c\} \tag{65}$$

## 6. Extension to Non-Sinusoidal Three-Phase Power Systems

The unbalanced apparent power in an unbalanced three-phase system according to [10] is only due to the fundamental components of the currents and voltages, whereas in a non-linear system, powers that are not due to the fundamental harmonics of current and voltage are considered harmonic apparent powers.

Consider a three-phase nonlinear system, where $V_{zm}$ is the per-phase voltage of harmonic order m and $I_{zn}$ is the per-phase current of harmonic order n where $z = \{a, b, c\}$. If we extend the vector expressions (28)–(30) that have been defined in previous sections for $m = n = 1$ to the remaining harmonic combinations of $m$ and $n$, that is, for each $m \neq n$ and $m = n \neq 1$, we obtain the harmonic vectors expressions according to (66)–(68).

$$\overrightarrow{S^n_{vm+}} = \sqrt{3}\left( \sum P^{zn}_{zm+} \overrightarrow{u^{zn}_{pzm+}} + \sum Q^{zn}_{zm+} \overrightarrow{u^{zn}_{qzm+}} \right) \tag{66}$$

$$\overrightarrow{S^n_{vm+}} = \sqrt{3}\left( \sum P^{zn}_{zm+} \overrightarrow{u^{zn}_{pzm+}} + \sum Q^{zn}_{zm+} \overrightarrow{u^{zn}_{qzm+}} \right) \tag{67}$$

$$\overrightarrow{S^n_{vm0}} = \sqrt{3}\left( \sum P^{zn}_{zm0} \overrightarrow{u^{zn}_{pzm0}} + \sum Q^{zn}_{zm0} \overrightarrow{u^{zn}_{qzm0}} \right) \tag{68}$$

$S^n_{vm+}$, $S^n_{vm-}$, and $S^n_{vm0}$ are the harmonic powers caused by the positive-, negative-, and zero-sequence voltages of order $m$ and the currents of order $n$. Its moduli are determined from the expressions (69)–(71).

$$S^n_{vm+} = \sqrt{3}\sqrt{\sum\left(P^{zn}_{zm+}\right)^2 + \sum\left(Q^{zn}_{zm+}\right)^2} \tag{69}$$

$$S^n_{vm-} = \sqrt{3}\sqrt{\sum\left(P^{zn}_{zm-}\right)^2 + \sum\left(Q^{zn}_{zm-}\right)^2} \tag{70}$$

$$S_{vm0}^n = \sqrt{3}\sqrt{\sum\left(P_{zm0}^{zn}\right)^2 + \sum\left(Q_{zm0}^{zn}\right)^2} \tag{71}$$

$\overrightarrow{u_{pzm+}^{zn}}, \overrightarrow{u_{qzm+}^{zn}}, \overrightarrow{u_{pzm-}^{zn}}, \overrightarrow{u_{qzm-}^{zn}}, \overrightarrow{u_{pzm0}^{zn}}$, and $\overrightarrow{u_{qzm0}^{zn}}$ are the unit vectors associated with each component. $P_{zm+}^{zn}, P_{zm-}^{zn}$, and $P_{zm0}^{zn}$ are the per-phase harmonic active powers caused by the positive-, negative-, and zero-sequence voltages, respectively, of order $m$ and the currents of order $n$. Their values are determined by (72), where $w = \{+, -, 0\}$.

$$P_{zmw}^{zn} = V_{zmw} I_{zn} \cos(\alpha_{zmw} - \beta_{zn}) \tag{72}$$

$Q_{zm+}^{zn}, Q_{zm-}^{zn}$, and $Q_{zm0}^{zn}$ are the per-phase harmonic reactive powers caused by the positive-, negative-, and zero-sequence voltages, respectively, of order $m$ and the currents of order $n$. Their values are determined by (73), where $w = \{+, -, 0\}$.

$$Q_{zmw}^{zn} = V_{zmw} I_{zn} \sin(\alpha_{zmw} - \beta_{zn}) \tag{73}$$

For each harmonic of voltage of order $m$ and harmonic of current of order $n$, the harmonic apparent power is given by (74).

$$S_m^n = \sqrt{\left(S_{vm+}^n\right)^2 + (S_{vm-}^n)^2 + \left(S_{vm0}^n\right)^2} \tag{74}$$

The total harmonic apparent power is given by

$$S_h = \sqrt{\sum_{\substack{m \neq n \\ m = n \neq 1}} S_m^{n\,2}} \tag{75}$$

Therefore, the total apparent power of the system $S$ is given by (76), and its value coincides with [10].

$$S = \sqrt{S_1{}^2 + S_h{}^2} \tag{76}$$

## 7. Practical Application

In this section, a practical case study to verify all of the concepts discussed in the previous sections is developed. Figure 7 shows a three-phase, four-wire electrical system with three unbalanced three-phase linear loads. The loads are modelled at a constant impedance. The impedance values of loads are listed in Tables 1–3. The voltages are unbalanced and sinusoidal in bus 1 (slack bus), in which

$$V_{an} = 230\, e^{j0}, \; V_{bn} = 205\, e^{-j115}, \; V_{cn} = 220\, e^{j135}$$

Using the 'PSPICE' analysis software, the line-to-neutral voltages in the buses (1, 2, and 3) and the currents circulating in the loads are obtained. These magnitudes are displayed in Tables 4 and 5.

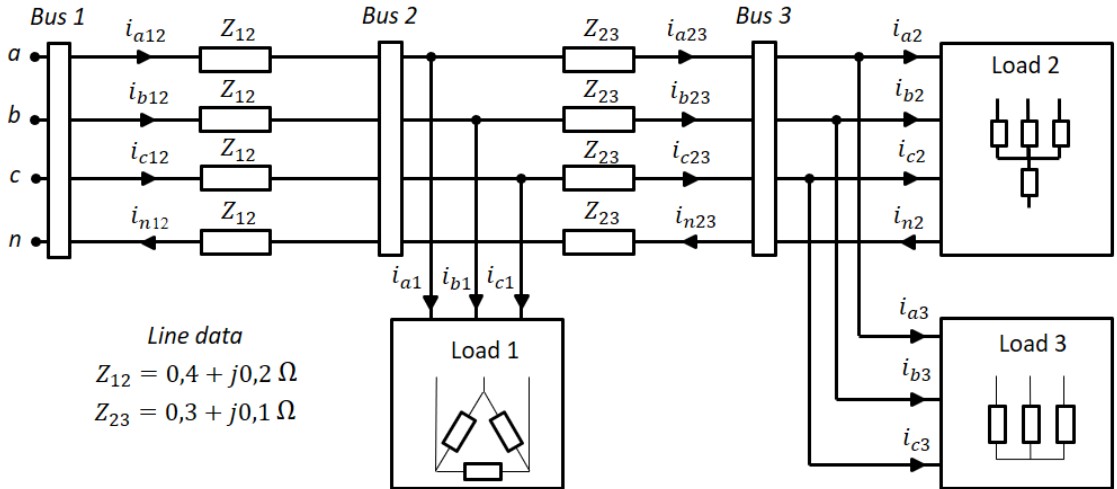

**Figure 7.** Electric system with unbalanced three-phase load.

**Table 1.** Load-1 Impedances.

|            | $R$      | $X$      |
| :---       | :---:    | :---:    |
| **Impedances** | **($\Omega$)** | **($\Omega$)** |
| $Z_{ab}$   | 12       | 3        |
| $Z_{bc}$   | 8        | 4        |
| $Z_{ca}$   | 5        | 2        |

**Table 2.** Load-2 Impedances.

|            | $R$      | $X$      |
| :---       | :---:    | :---:    |
| **Impedances** | **($\Omega$)** | **($\Omega$)** |
| $Z_a$      | 17       | 5        |
| $Z_b$      | 7        | 1        |
| $Z_c$      | 10       | 3        |
| $Z_n$      | 0.2      | 0.1      |

**Table 3.** Load-3 Impedances.

|            | $R$      | $X$      |
| :---       | :---:    | :---:    |
| **Impedances** | **($\Omega$)** | **($\Omega$)** |
| $Z_a$      | 12       | 3        |
| $Z_b$      | 8        | 4        |
| $Z_c$      | 5        | 2        |

**Table 4.** Line-to-neutral voltage.

|        | $V_{an}$ (V) | | $V_{bn}$ (V) | | $V_{cn}$ (V) | |
| :---   | :---:   | :---:   | :---:   | :---:    | :---:   | :---:   |
|        | **Modulus** | **Modulus** | **Modulus** | **Modulus** | **Modulus** | **Angle** |
| Bus 2  | 187.88  | 0.49    | 161.97  | −116.90  | 171.72  | 132.68  |
| Bus 3  | 180.50  | -0.09   | 143.48  | −120.47  | 161.03  | 130.36  |

**Table 5.** Line currents of the loads.

| | $I_a$ (A) | | $I_b$ (A) | | $I_c$ (A) | |
| | Modulus | Modulus | Modulus | Modulus | Modulus | Angle |
|---|---|---|---|---|---|---|
| Load 1 | 73.314 | −28.268 | 48.219 | −132.487 | 79.632 | 115.789 |
| Load 2 | 10.259 | −15.942 | 19.990 | −128.500 | 15.496 | 112.913 |
| Load 3 | 14.334 | −8.586 | 20.070 | −136.064 | 16.068 | 89.005 |

Table 6 shows the voltages expressed in symmetric components for each of the nodes of the system. These values are referred to as phase A.

**Table 6.** Positive-, negative-, and zero-sequence line-to-neutral voltage.

| | $V_{a+}$ (V) | | $V_{a-}$ (V) | | $V_{a0}$ (V) | |
| | Modulus | Modulus | Modulus | Modulus | Modulus | Angle |
|---|---|---|---|---|---|---|
| Bus 1 | 217.012 | 6.598 | 23.722 | −38.783 | 10.866 | −111.978 |
| Bus 2 | 173.122 | 5.308 | 18.389 | −28.891 | 5.568 | −96.214 |
| Bus 3 | 161.054 | 3.260 | 20.632 | −25.965 | 1.223 | −19.535 |

Table 7 shows the values of the components that form the vectors of the apparent power and unbalanced power, considering the positive-sequence voltage. To calculate $P_{z+}^z$ and $Q_{z+}^z$, expressions (16) and (17) have been used. For $P_+$ and $Q_+$, expressions (33) and (34) have been used.

**Table 7.** Components of the apparent power vector and unbalanced power due to positive-sequence voltage.

| | Phase A | | Phase B | | Phase C | | $S_+$ | |
| | $P_{a+}^a$ | $Q_{a+}^a$ | $P_{b+}^b$ | $Q_{b+}^b$ | $P_{c+}^c$ | $Q_{c+}^c$ | $P_+$ | $Q_+$ |
|---|---|---|---|---|---|---|---|---|
| Load 1 | 11,077.32 | 7306.69 | 7948.36 | 2551.11 | 13,596.30 | 2279.77 | 32,551.98 | 12,137.58 |
| Load 2 | 1560.41 | 543.46 | 3151.92 | 656.18 | 2450.88 | 447.45 | 7163.20 | 1647.09 |
| Load 3 | 2259.38 | 473.88 | 3050.17 | 1069.62 | 2138.86 | 1456.60 | 7448.41 | 3000.10 |
| Line 1-2 | 3574.94 | 2589.27 | 3551.66 | 1586.16 | 4545.42 | 1660.58 | 11,672.02 | 5836.01 |
| Line 2-3 | 244.53 | 222.28 | 394.20 | 366.40 | 267.63 | 317.69 | 906.36 | 906.36 |
| Bus 1 | 18,646.58 | 11,135.59 | 18,096.31 | 6229.46 | 22,999.08 | 6162.09 | 59,741.97 | 23,527.13 |
| Bus 2 | 15,071.64 | 8546.31 | 14,544.65 | 4643.30 | 18,453.66 | 4501.51 | 48,069.95 | 17,691.13 |
| Bus 3 | 3819.79 | 1017.34 | 6202.09 | 1725.79 | 4589.73 | 1904.05 | 14,611.61 | 4647.19 |

Table 8 shows the values of the components that form the vectors of apparent power and unbalanced power, considering the negative-sequence voltage. To calculate $P_{z-}^z$ and $Q_{z-}^z$, expressions (24) and (25) have been used.

**Table 8.** Components of the apparent power vector and unbalanced power due to negative-sequence voltage.

| | Phase A | | Phase B | | Phase C | |
| | $P_{a-}^a$ | $Q_{a-}^a$ | $P_{b-}^b$ | $Q_{b-}^b$ | $P_{c-}^c$ | $Q_{c-}^c$ |
|---|---|---|---|---|---|---|
| Load 1 | 1403.25 | −15.26 | −642.17 | −611.42 | −135.75 | 1458.04 |
| Load 2 | 208.44 | −36.84 | −303.91 | −278.82 | −61.56 | 313.17 |
| Load 3 | 282.24 | −88.34 | −265.61 | −317.66 | −190.29 | 271.45 |
| Line 1-2 | 463.83 | −466.88 | −566.03 | −41.73 | 350.70 | 612.86 |
| Line 2-3 | −59.61 | −8.57 | 35.44 | 91.45 | 54.27 | −52.78 |
| Bus 1 | 2298.15 | −595.88 | −1742.28 | −1158.19 | 17.37 | 2602.74 |
| Bus 2 | 1834.32 | −149.01 | −1176.25 | −1116.45 | −333.33 | 1989.88 |
| Bus 3 | 490.68 | −125.18 | −569.53 | −596.48 | −251.85 | 584.62 |

Table 9 shows the values of the components that form the vectors of apparent power and unbalanced power, considering the zero-sequence voltage. To calculate $P_{z0}^z$ and $Q_{z0}^z$, expressions (26) and (27) have been used.

**Table 9.** Components of the apparent power vector and unbalanced power due to zero-sequence voltage.

|          | Phase A | | Phase B | | Phase C | |
|----------|---------|---------|---------|---------|---------|---------|
|          | $P_{a0}^a$ | $Q_{a0}^a$ | $P_{b0}^b$ | $Q_{b0}^b$ | $P_{c0}^c$ | $Q_{c0}^c$ |
| Load 1   | 159.55  | −393.83   | 216.45  | 158.84  | −376.01 | 234.98  |
| Load 2   | 12.52   | −0.79     | −7.95   | 23.12   | −12.77  | −13.96  |
| Load 3   | 17.21   | −3.33     | −10.96  | 21.96   | −6.25   | −18.63  |
| Line 1-2 | −129.46 | −556.77   | 501.75  | 44.43   | −323.36 | 536.80  |
| Line 2-3 | −16.78  | −131.93   | 198.80  | 85.98   | −145.32 | 82.65   |
| Bus 1    | 43.05   | −1086.64  | 898.09  | 334.34  | −863.70 | 821.85  |
| Bus 2    | 172.51  | −529.87   | 396.34  | 289.90  | −540.34 | 285.05  |
| Bus 3    | 29.74   | −4.12     | −18.91  | 45.08   | −19.02  | −32.59  |

Finally, Tables 10 and 11 show the moduli of the apparent power vectors and unbalanced power vectors, respectively.

**Table 10.** Results of apparent powers.

|          | $S_{v+}$ | $S_{v-}$ | $S_{v0}$ | $S_1$ |
|----------|----------|----------|----------|----------|
|          | Modulus  | Modulus  | Modulus  | Modulus  |
| Load 1   | 36,095.29 | 3834.01 | 1160.91 | 36,316.91 |
| Load 2   | 7609.68  | 974.85   | 57.79    | 7672.08  |
| Load 3   | 8210.98  | 1051.88  | 62.35    | 8278.31  |
| Line 1-2 | 13,194.66 | 1925.28 | 1708.68 | 13,443.41 |
| Line 2-3 | 1309.27  | 238.60   | 526.92   | 1431.35  |
| Bus 1    | 64,920.90 | 7096.78 | 3250.72 | 65,388.49 |
| Bus 2    | 51,790.99 | 5501.19 | 1665.72 | 52,108.97 |
| Bus 3    | 15,661.52 | 2006.34 | 118.93   | 15,789.96 |

**Table 11.** Results of unbalanced powers.

|          | $S_{v+}$ | $S_{v-}$ | $S_{v0}$ | $S_1$ |
|----------|----------|----------|----------|----------|
|          | Modulus  | Modulus  | Modulus  | Modulus  |
| Load 1   | 9793.79  | 3834.01  | 1160.91  | 10,581.38 |
| Load 2   | 1970.48  | 974.85   | 57.79    | 2199.20  |
| Load 3   | 1714.86  | 1051.88  | 62.35    | 2012.73  |
| Line 1-2 | 1950.43  | 1925.28  | 1708.68  | 3229.62  |
| Line 2-3 | 266.83   | 238.60   | 526.92   | 637.01   |
| Bus 1    | 9596.99  | 7096.78  | 3250.72  | 12,370.37 |
| Bus 2    | 7655.73  | 5501.19  | 1665.72  | 9573.29  |
| Bus 3    | 3191.83  | 2006.34  | 118.93   | 3771.91  |

## 8. Conclusions

In this study, a different representation of the expressions for the unbalanced power vectors and apparent power vectors of a linear three-phase electrical system unbalanced in currents and voltages has been proposed. It was verified that the modulus of apparent power and unbalanced power is the same as that obtained by the Buchholz method. The proposed vectors allow the evaluation and determination of the unbalanced power anywhere in an electrical system, without requiring the use of equivalent circuits at the nodes of the system. Using the proposed vectors, it has been shown that it is sufficient to calculate the voltages at nodes and line currents at loads to determine the unbalanced

power flows. Each of these vectors contains six components, distributed in two components per phase that are expressed in terms of active and reactive powers. It has been proven that in the assumption of two loads connected in parallel, through the components of the vectors of each one, the resulting vectors of the group of loads are easily determined, regardless of the characteristics and type of connection of the loads. This allows us to quickly see if the unbalance power will increase or decrease by coupling the loads in parallel at the connection bus. This analysis cannot be performed directly from the currents and voltages, mainly when the voltages are unbalanced. Similarly, this advantage extends to the analysis of the unbalance power flows between various buses in the system to correctly assess the losses in the lines. To validate the applicability of the proposed vectors and improve comprehension, a practical case of a three-phase four-wire linear system with three buses was developed and studied.

**Author Contributions:** Conceptualization, P.A.B., R.M., and J.M.D.; methodology, P.A.B., R.M.-M., J.M.D., and R.M.; validation, P.A.B., R.M.-M., and R.M.; formal analysis, P.A.B. and R.M.-M.; investigation, P.A.B., R.M.-M., J.M.D., and R.M.; resources, P.A.B. and R.M.-M.; data curation, P.A.B. and R.M.-M.; writing—original draft preparation, J.M.D. and R.M.; writing—review and editing, R.M.-M., J.M.D., and R.M.; visualization, P.A.B. and R.M.-M.; supervision, J.M.D. and R.M.; project administration, J.M.D. All authors have read and agreed to the published version of the manuscript.

**Funding:** This work is supported by the Spanish Ministry of Science, Innovation and Universities (MICINN) and the European Regional Development Fund (ERDF) under Grant RTI2018-100732-B-C21.

**Conflicts of Interest:** The authors declare no conflict of interest.

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
