# Peer review of "An Alternate Representation of the Vector of Apparent Power and Unbalanced Power in Three-Phase Electrical Systems"

_applsci, doi:10.3390/app10113756_

Round 1

Author Response

Dear Reviewer

Attached find review reply file.

Reviewer 2 Report

Major issues:
Manuscript presents different representation of the expressions for the unbalanced power vectors and apparent power vectors of a linear three-phase electrical system unbalanced in currents and voltages have been proposed.
Introduction: appropriate review of previous research is lacking. “….There are other studies that have proposed phasor or vector expressions to evaluate these inefficiencies [15-18]…” the scope and results of the study should be clearly indicated.
The goal of the study is not clearly stated. The last paragraph of the introduction should clearly define what is the purpose of research, and what is the novelty of this manuscript.
The scientific work should have a clear and legible layout. The obtained research results should be better presented in the paper. The conclusions should indicate whether the aim of the study was achieved.
Minor issues:
Two points have the same number:
Line 270: 7. Practical application
Line 315: 7. Conclusions.
Before the use, abbreviation needs to be defined, the full term needs to be stated: RMS.

Author Response

(The authors gave the same response as above.)

Reviewer 3 Report

Positive Points:

1) Literature survey

2) Presentation

3) English 

Concerns:

1) Similarity with authors' previous work to some extent

2) It has been pointed out in the paper that node level imbalance analysis can help in better compensation. But, the shown derivations seem to show no such additional knowledge as claimed. While the derivations are elaborate, it is not very clear wether the nodal level knowledge of imbalance can be ascertained with the proposition.

3) A significant portion of the derivations show the widely known equations and relations. Some expressions can be used directly by citing original sources. It is advisable to use the available length of paper, more towards explaning the unknown/ proposed rather than highlighting the known equations. The empahsis on leverage that the proposed technique offers in compensation, is not profound. 

4) Some references were missing in the main body of the paper. For example - [27]. Please verify the presence of all references at the right place.

5) Please use consistent system description throughout the paper, otherwise their could be possible errors. For example- at line 337 it is written 'four-wire three-wire linear system with two buses'. Did the authors mean 'four-wire three-phase'? Please also verify the number of buses - is it 2 or 3?

6) The paper mentions 'nonlinear systems' in the first half, and then talks about 'linear systems' with unbalanced loads and voltages in the later half. Kindly clarify what type of system would you like to validate your proposition on? Again, be consistent with terminology!

Author Response

(The authors gave the same response as above.)
